

# Assessing the sources of particles at an urban background site using both regulatory instruments and low-cost sensors – A comparative study

Dimitrios Bousiotis[1], Ajit Singh[1], Molly Haugen[3], David C.S. Beddows[1], Sebastián Diez[2], Pete M. Edwards[2], Adam Boies[3], Roy M. Harrison[1] and Francis D. Pope[1]

[1] Division of Environmental Health and Risk Management
School of Geography, Earth and Environmental Sciences
University of Birmingham, Edgbaston, Birmingham B15 2TT, United Kingdom

[2] Wolfson Atmospheric Chemistry Laboratories, Department of Chemistry, University of York, Heslington, York YO10 5DD, United Kingdom

[3] Department of Engineering, University of Cambridge, Trumpington Street, Cambridge, CB2 1PZ, United Kingdom



**Abstract**
Measurement and source apportionment of atmospheric pollutants is crucial for the
assessment of air quality and the implementation of policies for its improvement. In most
cases, such measurements use expensive regulatory grade instruments, which makes it
difficult to achieve wide spatial coverage. Low-cost sensors may provide a more affordable
alternative, but their capability and reliability in separating distinct sources of particles have
not yet been tested extensively. The present study examines the ability of a low-cost Optical
Particle Counter (OPC) to identify the sources of particles and conditions that affect particle
concentrations at an urban background site in Birmingham, UK. To help evaluate the results,
the same analysis is performed on data from a regulatory-grade instrument (SMPS) and
compared to the outcomes from the OPC analysis. The analysis of the low-cost sensor data
manages to separate time periods and atmospheric conditions according to the level of
pollution at the site. It also successfully identifies a number of particle sources, which were
also identified using the regulatory-grade instruments. The low-cost sensor, due to the
particle size range measured (0.35 to 40 μm), performed rather well in differentiating sources
of particles with sizes greater than 1 μm. However, the ability of the low cost sensor to
distinguish diurnal variations and separate sources of smaller particles was more limited. This
study highlights the current capability of low-cost sensors in source identification and
differentiation using clustering approaches. The current level of source identification
demonstrated in this paper indicates the combination of hardware and analytical technique
is useful for background site studies, where larger particles with smaller temporal variations
are of significant importance. Future directions towards particulate matter source
apportionment using low cost OPCs are highlighted.



## 1. Introduction



Particulate matter (PM) plays a dominant role in air quality and is known to cause adverse
health effects (Dockery et al., 1993; Pascal et al., 2013; Wu et al., 2016; Zeger et al., 2008). As
a result, regulatory limits are set for its concentrations, especially in urban areas (US EPA,
2012; WHO, 2006). For the implementation of such regulations, the identification of the
sources of PM is required. To accomplish this, measurements of the concentrations of PM,
typically alongside PM composition, in the area of study are conducted. Until recent years
these measurements were usually made using regulatory-grade instruments which, while
providing high quality data, are rather expensive thereby limiting the number that could be
deployed and consequently the spatial resolution of any measurement network. This
increases the spatial interpolation uncertainty (Kanaroglou et al., 2005) and can result in
inadequate connection between the levels of air pollution exposures and health effects
(Holstius et al., 2014), especially in complex urban environments (Harrison, 2017; Mueller et
al., 2016). Additionally, many low and middle income countries are unable to invest the large
economic assets currently required for source apportionment, even though in many of these
countries, the air quality is poor (Ghosh and Parida, 2015; Kan et al., 2009; Petkova et al.,
2013; Pope et al., 2018; Singh et al., 2020).
In the past decade, the development of new and cheaper sensors for air quality monitoring
has intensified. Many different sensors were introduced measuring either the number
concentration or surface area of PM, or the gas phase species (Jovašević-Stojanović et al.,
2015; Lewis et al., 2018; Popoola et al., 2018). Overall, low-cost PM sensors currently offer
better comparison with regulatory grade equipment compared to their gas phase
counterparts (Lewis et al., 2018). However, many shortcomings have been identified in their
application, with the most common being the loss of measurement accuracy due to
environmental conditions such as relative humidity (RH) variations or high PM concentrations
(Castell et al., 2017; Crilley et al., 2018; 2020; Di Antonio et al., 2018; Miskell et al., 2017;
Zheng et al., 2018). Measurements in ambient conditions also lead to discrepancies with
research-grade instruments, which often measure in controlled environments that are air
conditioned (U.S. Environmental Protection Agency, 2016). The reproducibility and variability
of the outputs from sensors of the same type can also be problematic (Austin et al., 2015;



Sousan et al., 2016; Wang et al., 2015). Therefore, the need for constant and careful
calibration is repeatedly highlighted in many studies that evaluate the potential of low cost
sensors (Rai et al., 2017; Spinelle et al., 2015, 2017). When these calibration steps are
implemented, low-cost sensors have been shown to provide reliable near-real time
measurements, maintaining high correlations with research-grade instruments (Kelly et al.,
2017; Malings et al., 2020; Sayahi et al., 2019) with the added advantages of lower cost and
portability.
Consequently, low-cost sensors have been successfully deployed in many studies for which
the use of more expensive instruments was not feasible. There is a number of applications in
low and middle income countries (e.g. Nagendra et al., 2019; Pope et al., 2018), in studies
which included mobile measurements within the urban environment (Ionascu et al., 2018;
Jerrett et al., 2017; Miskell et al., 2018), or studies of indoor air quality from multiple sites,
such as the SKOMOBO project conducted in New Zealand, in which the air quality in schools
was assessed (Weyers et al., 2018). The greatest advantage though is likely, as their name
implies, their lower cost which made possible the formation of a network of measuring
stations (Feinberg et al., 2019; Kotsev et al., 2016; Moltchanov et al., 2015), increasing the
spatial resolution and through new data analysis methods improve the mapping of air
pollution up to a sub-neighbourhood level (Schneider et al., 2017). Therefore, it is suggested
that the development and use of low-cost sensors, either used individually or in conjunction
with research-grade instruments (Snyder et al., 2013), have the potential to radically change
the conventional approach of both pollution measuring and policy making (Borrego et al.,
2018; Kumar et al., 2015; Lagerspetz et al., 2019), providing a more effective general public
information and enhanced environmental awareness (Penza et al., 2014), even for countries
with smaller budgets (Amegah, 2018).
As yet, studying the different sources of particles at a site with the use of data from low-cost
sensors has not been widely attempted yet. Pope et al., (2018) managed to identify major
pollution sources studying the ratios of PM of different sizes provided by low-cost sensors,
while Popoola et al., (2018) using a network of sensors identified the sources of pollution near
Heathrow airport in London, UK. Hagan et al., (2019) applying a statistical method (Non-
negative Matrix Factorisation) on low-cost sensor data, identified a combustion factor in a
three-factor solution in New Delhi, India. The present study investigates the ability of low-
cost sensors to provide measurements that can be used to identify the sources of pollution





at a background site in Birmingham, UK, using clustering of particle composition profiles. This
method was successfully used in a number of previous studies, though with the use of
measurements from research-grade instruments (Beddows et al., 2009, 2015; Von Bismarck-
Osten and Weber, 2014; Dall'Osto et al., 2011; Sabaliauskas et al., 2013). To support the
clustering method, chemical composition data from both research-grade and low-cost sensor
instruments were used, as well as meteorological data from a closely located measurement
station. Apart from attempting the source differentiation with low-cost sensor data, a direct
comparison with the results from a similar analysis using research-grade instruments is also
conducted to not only validate the results but find the strengths and weaknesses of such an
application.

## 2. Methods


### 2.1 Location of the site and instruments


The measurement site (fig. 1), characterised as an urban background, is the Birmingham Air
Quality Supersite (BAQS) located at the grounds of the University of Birmingham (52.45°N;
1.93°W), about 3 km southwest from the city centre (Alam et al., 2015). In the present study,
measurements from the following instruments for the period 24/01/2020 to 12/3/2020 (the
date range was chosen to avoid the effect of the lockdown due to COVID-19) were used (Table

123    1):

The Alphasense OPC-N3, which is an optical particle counter, measuring particle number
concentrations in the size range between 0.35 to 40 μm at rates up to about 10000 particles
per second. As the sample air stream enters the instrument, it passes through a laser beam
and the particle size and number concentrations are derived from the light scattered by the
particles, based on the Mie scattering theory. It can also provide data for particle mass
loadings ($PM_1$ to $PM_{10}$) assuming a particle density, shape and refractive index. The OPC is
located within the air conditioned station, so measurements represent PM dry mass.
The AethLabs MA200 (microAeth MA200) which provides black carbon (BC) information (0-1
mg BC/$m^3$). The sample is deposited onto an internal filter, and an IR light (880 nm) is directed
through the sample on the filter and into a detector on the other side of the sample. The
amount of light absorbed from the sample is proportional to the BC concentration.



Two Naneos Partectors (Naneos Particle Solutions GmbH) which provide the lung deposited
surface area metric (LDSA, $\mu m^2/cm^3$) in the particle diameter range 10 nm to 10 µm. In
general, the provided data is dictated by the particle number concentration and diameter
($Nd^{1.1}$) for both semi-volatile and solid particles. A catalytic stripper (Catalytic Instruments
CS015) was used to remove the semi-volatile particles entering one of the two Naneos
Partectors. The other Naneos Partector was not subject to the catalytic stripper and therefore
measured the surface of all particles. In the present study, apart from the values provided
directly from the sensors, the ratio between the measurements of the two Naneos Partectors
was also considered according to:

$$LDSA_{ratio} = \frac{LDSA \; after \; the \; catalytic \; stripper}{LDSA \; before \; the \; catalytic \; stripper}$$

This was done to resolve whether such a configuration can also provide information such as
the level of pollution or the age of the incoming air masses, as increased concentrations of
semi-volatile compounds are usually associated with anthropogenic sources, especially in the
urban environment (Harkov, 1989; Schnelle-Kreis et al., 2007). Thus, a high $LDSA_{ratio}$ is
expected to be associated with fresher pollution (i.e., pollution sources at a close distance
from the site), while lower ratios are probably associated with either cleaner conditions or
more regional and aged pollution, usually associated with sources at a greater distance from
the measuring site. The specific metric though should be considered with caution, as it can be
biased by the absolute surface areas measured.
The sensors monitoring nitrogen dioxide ($NO_2$) and ozone ($O_3$) concentrations are part of a
Box Of Clustered Sensors (BOCS) (Smith et al., 2019), which is a low-power instrument based
on the clustering of multiple low-cost air pollution sensors allocated in two independent
circuits to redundantly measure concentrations and other airflow parameters. The air is
driven by a pump through the cell that hosts the electrochemical sensors (EC) and the
nondispersive infrared sensors (NDIR). While the EC sensors redundantly (6 sensors per gas)
measure carbon monoxide, $NO_2$, nitrogen monoxide, oxidizing gases ($O_x$), the NDIR sensors
measure carbon dioxide. EC sensors are based on recording the current generated by redox
reactions that occur at the electrode-electrolyte interface in an electrochemical cell
composed of three electrodes (working electrode (WE), counter electrode (CE) and reference





electrode (RE)). While the gas of interest reacts on the WE surface, the CE completes the
redox reaction and the RE ensures that the WE potential remains in the proper range. In the
present study, measurements of $O_x$ and $NO_2$ were only used from the specific sensor.
For the same period data from regulatory-grade instruments were also available. Thus,
particle size composition data from a Scanning Mobility Particle Sizer (SMPS) in the size range
12 – 552 nm, along with PM data for the sizes of 1, 2.5, 4 and 10 μm acquired using a Fidas
200E were used. Additionally, chemical composition data for $NO_2$, $O_3$, as well as $SO_4^{2-}$, $NO_3^-$
and organic content (size range 40 nm to 1 μm) from an Aerosol Chemical Speciation Monitor
(ACSM) were also available. Meteorological data (wind speed and direction, temperature, RH
and rain level) from the Birmingham Air Quality Supersite were also used in the
characterisation of the clusters formed from both methods.
Back trajectory data calculated using the HYSPLIT model (Draxler and Hess, 1998), were
extracted        by        the        NOAA        Air        Resources        Laboratory
(https://ready.arl.noaa.gov/READYtransp.php). Data was processed using the Openair
package for R (Carslaw and Ropkins, 2012).

**2.2 k-means clustering**
In this study, two size spectra are considered, one deriving from the OPC and one from the
regulatory-grade SMPS. It is noted that the size spectra from the two instruments only briefly
overlap in the size range 350 – 552 nm, with the SMPS mostly measuring smaller particles and
the OPC mostly measuring larger particles.  For the period studied (24/1/2020 – 12/3/2020),
874 hours of available data (averaged from 10 second intervals - 76% coverage) from the OPC
and 732 hours from the SMPS (66% coverage) were exposed to k-means clustering. k-means
clustering is a method successfully used in many studies of particle source differentiation
(Beddows et al., 2015; Von Bismarck-Osten and Weber, 2014; Giorio et al., 2015; Wegner et
al., 2012) and was proven to have better performance compared to other clustering
techniques (Beddows et al., 2009; Salimi et al., 2014), as it was found to produce clusters with
the highest similarity between their elements and the highest separation against the other
clusters formed (Hennig, 2007). The optimal number of clusters was chosen using two
metrics, the Dunn Index and the Silhouette width as proposed by Beddows et al., 2009. The
Dunn Index provides a measure of the ratio of the minimum and the maximum cluster. The



larger the Dunn Index the better separated are the clusters formed. The Silhouette width is a
measure of the similarity of the spectra within each cluster. In the present study the best
statistically fitted solution was chosen, though in source differentiation studies such a
solution may not always provide with the best separation of all the available sources. Using
the aforementioned statistical tests, a six-cluster solution was independently suggested for
both the OPC and SMPS datasets.


## 3. Results

### 3.1 General conditions, sources of particles and pollution at the site

Being an urban background, the site studied presents relatively low concentrations of most
pollutants (Table 2), without the effect of direct sources of pollution, such as traffic. Wind
rose and polar plots illustrating the conditions in the period studied are found in figure S1.
The main source of pollution lies on the north and northeast sectors, where the city centre is
located, as well as in the southern and eastern sectors where a populous residential area is
located. As a result, the main sources of $NO_2$ and BC as well as the smaller sized PM are
associated with easterly winds (this though is not reflected in particles observed in the SMPS
size range). For the $PM_{10}$ apart from the aforementioned, increased concentrations are also
found with southwestern winds likely associated with marine sources. Typical for the UK, the
average wind profile for the period consists mainly of western and southwestern winds
(McIntosh and Thom, 1969), reducing the effect of the pollution sources in the east of the
site. Finally, the secondary pollutants $NO_3^-$ and $SO_4^{2-}$ which are in most cases associated aged
pollution and long-distance transport, have less consistent profiles, though they both seem
to be mainly associated with southern wind directions.

### 3.2 Clustering of the OPC data

Due to the larger particle sizes measured by the OPC-N3, the differences in the cluster profiles
are mainly associated with the particle number concentrations and to a lesser extent on the
different peaks, which are less distinct due to the smaller variation found as particle diameter
increases.  The frequency of the clusters formed, and their diurnal occurrence is shown in



figure 2. The average particle size distribution spectra and wind roses for the clusters formed are found in figures S2 and S3.

The six clusters formed from the OPC data are:

**OPC.1**: A rather polluted group with the highest $NO_2$ concentrations and average secondary pollutants, PM and LDSA ratio. Its fresher polluted character is further confirmed using the SMPS data which showed higher than average particle concentrations for particles with diameter smaller than 50 nm. This group presents low average temperature, RH and slower than average southwestern winds, which is explained, to an extent, by the cluster being more frequent during night-time.

**OPC.2**: The second group refers mainly to a single midday event on 12/3/2020 with high-speed southwestern winds, high temperature and very low RH. On this day the concentrations of all the pollutants were rather low, though due to the high wind speeds (an increase in the wind speed is observed at the start of the occurrence of this cluster – at 10:00 AM - which affects the particle distribution profile as can be seen in Figure S4) the $PM_{10}$ were close to average (when $PM_1$ and $PM_{2.5}$ were rather low) indicating the stronger presence of coarser particles, possible of marine origin as shown by the back trajectories, a source with an increasing importance at larger size PM at this area (Harrison et al., 2004; Taiwo et al., 2014). This group presents the highest LDSA ratio, which is in agreement with the low concentrations of the secondary pollutants.

**OPC.3**: A group occurring mainly during some of the midday periods in January, with the lowest temperature and wind speed averages, as well as the highest average RH, containing both southwestern and southern winds. While the concentrations of the measured pollutants are close to average, high sulphate and ozone concentrations were found, with the former pointing to air masses with higher concentrations of aged pollutants. The LDSA ratio though, was found to be very high despite the higher concentrations of sulphate and nitrate. The near average $NO_2$ concentrations may point to the effect of a nearby pollution source that may resulted to the increased LDSA ratio found.

**OPC.4**: A group with low concentrations of $NO_2$, BC and PM, but close to average secondary pollutants' concentrations. It is associated with close to average temperature, RH and wind speed of mainly southwestern directions. It is slightly more frequent during daytime and has lower than average concentrations of particles in the SMPS range.


**OPC.5**: This group includes the most polluted conditions in the area throughout the day. It is
associated with western and southwestern winds of average speed, high temperature and
lower than average RH. Most pollutant concentrations, including PM, are rather high while
$O_3$ is low. Similarly, it presents the highest concentrations of particles in all SMPS size ranges.
This cluster also includes the more polluted conditions found with north-eastern winds.
**OPC.6**: A group associated with rather clean conditions, presenting the lowest concentrations
of $NO_2$, BC, $NO_3^-$ and organic content. It is associated with higher than average temperature
and wind speed and lower than average RH, and has low concentrations of $PM_1$ and $PM_{2.5}$,
while $PM_{10}$ concentration is close to average. It is more frequent during daytime, which
probably explains the highest $O_3$ concentrations. The fast-moving southwestern air masses,
which this group is associated with, are probably of marine origin that have not passed
through any significant pollution sources, which can be further suggested by both the low
LDSA values and the highest LDSA ratio.

## 272    3.3 Clustering of the SMPS data

In the past, a number of studies on the sources of particles were conducted for both the
greater area of Birmingham and specifically the site in the University (Harrison et al., 1997;
Taiwo, 2016; Yin et al., 2010). As, these studies mainly focused on the chemical composition
of coarser particles, to the authors' knowledge this is the first study that uses ultrafine particle
size distribution data to study the sources of particles in Birmingham, UK. The frequency and
hourly occurrence of the six clusters formed from the SMPS data is found in figure 3. The
average particle size distributions and wind roses for the clusters formed are found in figures
S5 and S6.
**SMPS.1**: This group contains averagely polluted hours and is associated with fresher
pollutants (such as $NO_2$ or NO) and PM, while secondary pollutants such as $NH_4^+$, $NO_3^-$ and
$SO_4^{2-}$ are relatively low. Due to being associated with fresher emissions this group presents
higher than average concentrations of particles below 50 nm and a low LDSA ratio. It is
associated with average speed southwestern winds (it also includes the small portion of
north-eastern winds) and temperature, higher than average RH and occurs more frequently
during late night and early morning hours.



**SMPS.2**: Similar to the first group, average pollutants' concentrations are found in this group
with low concentrations of secondary pollutants. It is associated with slow western and
southwestern winds, lower than average temperatures and RH and is more frequent during
early morning hours. It has the highest concentrations of particles with diameter smaller than
20 nm, but the particle concentrations become relatively smaller as their size increase.
**SMPS.3**: This is a small group containing very clean night hours mainly in February, with higher
than average temperature, lower than average RH and strong western and southwestern
winds. It has low concentrations of pollutants and PM apart from $O_3$ (despite the time of day),
though $PM_{10}$ concentration is enhanced, probably associating this group with stronger marine
origins. The particle concentrations of all size ranges below 500 nm are the lowest among the
groups formed and along with the high LDSA ratio are in agreement with the very clean
conditions fassociated with this cluster. This cluster, contrary to all other, presents two peaks:
one peaking just below 30 nm and another one just over 100 nm, which indicates that it is
probably associated with at least two different sources.
**SMPS.4**: This group presents near average concentrations of all the pollutants studied. $PM_1$
average concentration is rather low while $PM_{10}$ is higher than the average. It is associated
with average speed southwestern winds, higher average temperature and low RH. It is more
frequent during midday and evening hours and it appears to represent the most common
conditions in the area, hence having the highest frequency of all clusters.
**SMPS.5**: This is a unique group associated with southern winds, the side at which the central
part of the University resides. This is the most polluted group, probably affected by emissions
from the University and the residential area found in that direction, with very high
concentrations of all the pollutants (apart from $O_3$), PM and ultrafine particles with available
data. The LDSA ratio is very high and this is probably due to the great surface area of the
involatile component found. It is associated with very slow wind speeds, low temperature,
very high RH and occurred evenly throughout the day, mainly on the first weeks of the
campaign when pollution levels were rather high, probably due to increased heating
emissions.
**SMPS.6**: This group presents low concentrations of all pollutants (apart from $O_3$), PM and
ultrafine particles with available data and is associated with western winds with higher than
average speed, near average temperatures and low RH. It occured more frequently during
evening hours and almost equally frequently throughout the whole study period apart from
the first 2 weeks when pollution levels were rather high.

**3.4 Direct comparison between the methods**

Due to the difference in the size ranges measured by the SMPS and OPC instruments, it is
evident that a direct comparison between the two methods would provide mixed results as
some clusters found using the SMPS data are not detectable with the OPC, and vice versa.
The particle size range that is common between the two instruments lies at about $350 - 550$
nm. Therefore, many particle sources associated with particles in the size range below the
minimum detectable size of the OPC are not expected to be found using its data and vice
versa. At a background site though, many of the sources of smaller sized particles play a less
important role as they are usually associated with fresher emissions, which are not common
to such sites.
The clustering process attempts to separate the particle size distributions into groups with as
similar spectral profiles as possible, while being as different to the other groups as possible.
As expected, the SMPS is more capable in separating different cluster profiles at the size range
smaller than 500 nm, a size range in which the cluster profiles formed by the OPC are almost
uniform (fig. 4). This shows the limitation of the OPC data to distinguish ultrafine particle
variations and thus it does not provide insight for the sources of particles within this size
range. On the other hand, the OPC performs much better in identifying different sources
when considering larger particles in the range between $1 - 10$ μm, for which it manages to
clearly distinguish variations between the groups formed (fig. 5). The clusters formed using
the OPC data were also better associated with different sources of $PM_1$ (fig. 6), compared to
those deriving from the SMPS data (fig. S7).
Table 3 contains the cluster relationships between the two methods, while Table S1 contains
the conditions observed when pairs of clusters from both methods are considered. The OPC.2
and OPC.3 clusters appear infrequently, and it would be nonsensical to directly associate
them with SMPS groups, as they appear under very specific conditions, that either are not
detected or are not identified as separate cases by the SMPS. As a result, they will be
separately studied later in this study.



The OPC.1 was mainly associated with SMPS.4 and SMPS.6 and to a lesser extend with
SMPS.1. OPC.1 has higher frequency during night times and it shares many of these hours
with groups SMPS.4 and SMPS.6, while with SMPS.1 it mainly shares early morning hours. It
includes the more polluted portion of the rather clean SMPS.6 and a portion with lower $PM_{10}$
(though not much difference from average pollutants' concentrations) from the more
polluted SMPS.4. It is interesting that the variation between the subgroups (in relation to
SMPS clusters) of the OPC.1 is very small for the $NO_2$ concentrations, a pollutant for which its
variations are not expected to be directly "visible" at the size range of the OPC as it is mainly
associated to fresher emissions. No great variation was found for the wind direction in the
subgroups of OPC.1, though it includes the lower temperature and higher RH conditions of
the SMPS clusters it is associated with. The OPC.1 includes the relatively clean part of the
more polluted SMPS.1 and the more polluted portion of the cleaner SMPS.6. While this does
not provide a clear connection between the OPC and SMPS results, it shows that there is
consistency in the results provided by the former in identifying particle sources of specific
qualities.
Similarly, OPC.4 was mainly associated with SMPS.4 and SMPS.6. As the OPC.4 occurs under
cleaner conditions, it includes the less polluted hours of both the SMPS clusters it is mainly
associated with, though the concentrations of the secondary pollutants such as $NO_3^-$ and $SO_4^{2-}$
are closer to the average. The OPC.4 is associated with the cleaner portion of the
aforementioned SMPS clusters with higher average temperature and RH though with variable
wind speeds.
OPC.5 represents a polluted group of hours associated mainly with SMPS.4, SMPS.5 and
SMPS.6. Being a group of hours associated with higher concentrations of pollutants, it
includes the more polluted portions of SMPS.4 and SMPS.6 with average meteorological
conditions, though lower wind speeds. It also coincides with the largest portion of SMPS.5,
mainly in the sixth week when the temperature was the lowest, including the portion with
the higher concentrations organic content and $NO_3^-$. SMPS.5 is the group that is associated
with southern wind directions, a side from which a source of secondary pollutants ($NO_3^-$, $SO_4^{2-}$
, $NH_4^+$), organic content and particles of diameter greater than 100 nm occurs. The OPC.5 is
associated with the part of SMPS.5 which is more burdened from secondary pollutants, hence
very large concentrations are observed for them.



Finally, OPC.6 is mainly associated with SMPS.2, SMPS.4 and SMPS.6. Being a cleaner group
of hours, it includes the portion of these SMPS clusters with lower pollutant concentrations
but higher $PM_{10}$ concentrations (though with lower $PM_1$ concentrations). These rather clean
conditions, along with the western and southwestern high-speed winds in average and the
large $PM_{10}$ concentrations, further enhance the possible marine character of this cluster. Due
to the size range of these particles such variation is not clearly identified by the SMPS,
resulting to them not being clearly separated when its data is considered.
The weekly contribution of each cluster group from the analysis of either dataset is found in
Figure 7 and the conditions on each week studied in Table S2. It is evident that the variation
from the SMPS is greater than that of the OPC, as the latter is less affected by the diurnal
variations. It is apparent that it is easier to comprehend the clusters' variation in association
with the levels of pollution in the site (the more polluted weeks have a greater portion of
SMPS.1 and SMPS.5), while for those with lower concentrations of pollutants the SMPS.4 and
SMPS.6 are more enhanced. These variations are harder to distinguish using the OPC data, as
they are less apparent in the size range measured by the sensor. To further understand the
possible sources using the latter, information from other instrument which provide chemical
composition data are needed, though it is still hard to pinpoint exact sources, due to the OPC's
weakness in explaining distinct particle sources within the day.

**3.5 Case studies**

**OPC.2**
OPC.2 occurs mainly on a single day in May (12[th]) with higher than average temperature and
strong western winds. This was the cluster with the lowest concentrations of $NH_4^+$, $NO_3$ (about
an order of magnitude compared to average conditions) and $SO_4^{2-}$, rather low concentrations
of $NO_2$, BC and high $O_3$. Using the SMPS data, this group of hours seems to follow the trends
of BC, associating it with SMPS.6 for low, SMPS.1 and SMPS.2 for medium and SMPS.4 for
higher concentrations of BC. This cluster has very low $PM_1$ and $PM_{2.5}$ and near average $PM_{10}$
concentrations, probably associating it with marine sources (due to the high wind speed). Due
to this, it is not clearly separated using the SMPS data, which does so for the hours of this
group according to the level of fresher pollutants, the variation of which is smaller in this type





412 of environments. This cluster seems to be the result of the change in the wind profile which

413 greatly affected the coarser particles at the site (figure S4).


415 **OPC.3**

416 The third cluster formed using the OPC data, was a rather small group of hours in late January

417 (25,27 and 28th), with the lowest average temperature and wind speed compared to the rest

418 of the clusters. The wind direction profile for this group contains both western and southern

419 winds, the latter being associated with high concentrations of pollutants (as found by the

420 study of the SMPS data). The majority of the hours in this group (65%) were characterised as

421 freshly polluted when using the SMPS data, mainly associated with SMPS.2. Unfortunately,

422 data of $NO_2$, BC, $O_3$ and PM for this group were very scarce from regulatory-grade instruments

423 (due to instrument error). The ACSM data, which were available for the hours of this cluster

424 pointed to marginally lower than average values of organic content, nitrate and ammonium,

425 while the sulphate concentrations were rather high. Using the low-cost sensor data, it is found

426 that this group has the highest BC, $O_3$ and involatile component of LDSA while $NO_2$, and CO

427 were the lowest among the groups. This group also had the highest average particle

428 concentration in the size range of the OPC, which is in agreement with the highest PM

429 concentrations in all ranges ($PM_1$, $PM_{2.5}$, $PM_{10}$). As this is not visible from the SMPS, the cluster

430 associated with this group has nearly average particle concentrations in the SMPS particle

431 ranges. This group was not distinctively detected by the SMPS due to presenting variation in

432 larger sized particles, which can be one of the weaknesses of studying the sources of such

433 particles using SMPS data alone. The OPC.3 appears to contain the more polluted slow-

434 moving portion of SMPS.2 with enhanced $SO_4^{2-}$, BC and PM concentrations.

435

436 **SMPS.3**

437 The third cluster from the analysis of SMPS data presented a unique profile with two peaks,

438 one below 30 nm and one a bit over 100 nm. This unique group was associated with very

439 clean conditions, with very low concentrations for all the pollutants with available data (apart

440 from $O_3$), as well as low particle concentrations for all the ranges in the SMPS and OPC range

441 as well as $PM_1$ and $PM_{2.5}$. The concentrations of $PM_{10}$ and $SO_4^{2-}$ were somehow higher but

442 still lower than the average in the area for the period of the study. This group is associated





with high average temperature and wind speed and rather low RH, with wind directions being
mainly southwestern and western. This group occurred solely at night hours during a number
of relatively warm nights mainly in February and to a lesser extend in March. Even with very
low particle concentrations (as found by both the SMPS and OPC) the presence of two
separate peaks in the size range of the ultrafine particles is indicative of more than one
simultaneous source. Due to these sources of particles occurring at the ultrafine particle
range, the OPC was not able to distinguish this special condition and grouped the hours of
this cluster to a number of clusters (mainly OPC.5 and to a lesser extend OPC.1 and OPC.6),
occurring either during night-time or throughout the day. The inability of the OPC to
distinguish complicated conditions in the ultrafine range is a weakness of the OPC that should
be considered when such conditions are anticipated.

## 4. Discussion


As the SMPS measures smaller particle sizes and with better accuracy, compared to the OPC,
it managed to better separate the different sources of fresher pollution with the main
differentiating factor being the time of the day, for which the variability of such sources is
more prominent. The differences in $NO_2$ concentrations, which are mainly associated with
fresher emissions are more distinct between the groups and using this data better separation
of very clean (SMPS.3) and very polluted conditions from a distinct source (SMPS.5) was
achieved, while the other groups described mostly average conditions with lesser variability
(as expected in this range at a background site). Additionally, using the SMPS data it is possible
to distinguish multiple sources of ultrafine particles (SMPS.3), as they can appear as multiple
peaks within the SMPS spectra. This is not possible using the OPC data as the size range
measured by this instrument cannot identify such cases.
Contrary to the SMPS, using the OPC data provided less distinct separation of fresher
emissions (as expected due to the lack of data of small sized particles). Additionally, the OPC
data is less sensitive to diurnal variations due to the range of particles covered, which are in
a size range that does not vary significantly through the day but between days. This results in
the less distinct diurnal variations found between the groups formed.  The analysis of the OPC
data though managed to adequately separate conditions and/or sources associated with





larger particles, such as aged pollution (for which it also managed to separate a small time-
window with very strong sulphate presence – OPC.3) which has the greatest contribution in
the particle chemical composition for the study area (Harrison et al., 2003; Taiwo, 2016; Yin
et al., 2010), RH variations or air masses of marine origin. To an extent, this might be due to
the number of clusters chosen as there is a possibility that a larger number of clusters from
the SMPS may separate sources of larger particles better, though with the risk of also
separating similar sources.
To sum up, the study of SMPS data with k-means clustering is far superior at separating
complex pollution sources within urban environments in which the variation of very small
particles is crucial for identifying particle and emission sources. This advantage of the SMPS
will not be overcome even with a denser measuring network of OPCs that could be acquired
for the same cost of the SMPS. However, clustering of the OPC data can provide useful
information to assess the sources of air pollution at background sites in which the direct
(smaller) particle sources are few. The method managed to find sources of greater pollution
associated with higher concentrations of particles of greater sizes (which are mainly
associated with aged pollution though), showing that the footprint of pollution in the ultrafine
particle range can have a detectable effect in coarser particle distributions as well. While not
as precise as the SMPS, a denser network of such instruments in background sites can be
more beneficial and more cost efficient in studying multiple pollution sources or "hot spots"
within the urban environment.
The current inability of low-cost PM sensors in measuring particle size spectra at small sizes
(<300 nm) is the greatest drawback in their application for separating particle sources, since
much information is contained in these smaller sizes.  OPCs using shorter wavelength light
sources and hence smaller particle detection could be beneficial here.  Also, there are several
low-cost sensors that provide insight for the surface area or the total number of particles in
the ultrafine particle size range (such as the LDSA sensor used in this study). The combined
use of the OPC with these instruments, along with sophisticated statistical techniques, may
provide possibilities for more precise source differentiation than shown in the present study.
It is noted that while clustering of particle number size distributions is one approach in the
study of the source assessment of particles, other alternative methods, such as the Positive
Matrix Factorisation (PMF), may also provide useful results.




## 5. Conclusions


The present study investigates the capabilities of a low-cost OPC sensor for source
differentiation at an urban background site in Birmingham, UK. It is used alongside a
regulatory-grade SMPS instrument, which has previously been used successfully for source
differentiation.  The clustering approach identified optimal solutions of six clusters for both
the SMPS and OPC data. There were similarities between the SMPS and OPC solutions, which
provide insights into periods of low and high pollution. However, large differences were also
observed. A more distinct separation of direct emission sources was achieved using the SMPS
data, which identified sources with time windows that correlated with extreme $NO_2$
concentrations (either high or low), as well as periods with more complex sources. The OPC
was able to distinguish time periods with greater variation of super micron sized particle
sources (e.g. marine sources). There seems to be a clearer distinction of the diurnal variability
of sources using the SMPS data, while the OPC seems to be able to only distinguish the
variability within periods of days rather than hours, as found by the less variable diurnal and
weekly variation. This though might not be a great drawback when considering background
sites, as this variability is smaller in such environments which are mainly affected by regional
pollution, while the local emissions are less and more distinct. Low-cost sensors can be a
reliable alternative for source identification studies in environments with less complex
sources, which present smaller alterations within the span of the day. Still, such instruments
cannot be used for scientific analyses which require greater precision. Their application will
probably be adequate when studying the sources of particles with a more regional character
(e.g. marine sources) rather than direct and variable sources (e.g. traffic or cooking emissions)
and can provide enough information for the air quality levels, sources and conditions these
are anticipated from. Such studies may include the analysis of mineral dust events resulting
from either anthropogenic activities or meteorological events (e.g. dust storms), bioaerosol
events in forested areas and other sources which affect mainly the composition of coarser
particles.
This study demonstrates that single low-cost sensor PM units can provide sensible source
differentiation of large sized PM pollution sources. This allows for the prospect of source



apportionment via networks of low-cost sensors in the near future, thereby allowing
triangulation of sources. The development of more sophisticated low-cost sensors in
conjunction with their low cost ensures the prospect of the application of a denser
measurement network, making better air quality monitoring and control feasible in the near
future. This though, requires more similar studies which can further elucidate the strengths
and weaknesses of those sensors compared to the regulatory-grade ones, as they develop.

**Author Contributions**
The study was conceived and planned by FDP who also contributed to the final manuscript,
and DB who also carried out the analysis and prepared the first draft. AS, MH, DCSB and SD
have provided with the data for the analysis. DCSB provided help with the analysis of the
data. RMH provided advice on the analysis. PME and AB contributed to the final manuscript.

**Competing Interests**
The authors have no conflict of interests.

**Acknowledgements**
The work is funded by NERC (NE/T001879/1) and EPSRC (EP/T030100/1). We thank the OSCA
team at the Birmingham Air Quality Supersite (BAQS), funded by NERC (NE/T001909/1), for
help in data collection for the regulatory grade instruments. We thank Lee Chapman for
access to his meteorological data set used in the analysis.



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


**TABLE LEGENDS**

**Table 1:**     List of the measuring instrument used in the present study.

**Table 2:**     Cluster conditions for both methods.

**Table 3:**     Cluster relationships between OPC and SMPS clusters.


**FIGURE LEGENDS**

**Figure 1:**     Map of the site (Map by ©HERE).

**Figure 2:**     Frequency and diurnal occurrence of the clusters formed by the OPC data.

**Figure 3:**     Frequency and diurnal occurrence of the clusters formed by the SMPS data.

**Figure 4:**     Particle contributions in the range 12 – 550 nm, for the clusters formed using

800             the OPC data (top) and the SMPS data (bottom).


**Figure 5:**     Particle contributions in the range up to 10 μm, for the clusters formed using

803             the OPC data (top) and the SMPS data (bottom).


**Figure 6:**     $PM_1$ polar plots of the clusters formed by the OPC data.

**Figure 7:**     Cluster percentage contribution per week (week number refers to week of

808             year 2020).




**Table 1: List of the measuring instrument used in the present study.**

| Monitoring | Model | Manufacturer | Regulatory grade | Approximate cost (£) |
|---|---|---|---|---|
| $NO_2$ | NO2-B43F | Alphasense | No | 250 |
| $O_x$ | Ox-B43I | Alphasense | No | 160 |
| Black Carbon | MA200 | Aethlabs | No | 5,700 |
| Lung Deposited Surface Area | | Naneos | No | 8,500 |
| OPC | OPC-N3 | Alphasense | No | 250 |
| SMPS | TSi3082 | TSi | Yes | 80,000 |
| ACSM | Quad - ACSM | Aerodyne | Yes | 170,000 |
| PM | Fidas 200E | Palas | Yes | 25,000 |
| $NO_2$ | T500U | Teledyne | Yes | 15,000 |
| Black Carbon | AE33 Aethalometer | Magee Scientific | Yes | 25,000 |
| $O_3$ | 49i | Thermo | Yes | 3,000 |







**Table 2: Cluster conditions for both methods.**

| | NO$_2$ (ppb) | BC (ng m$^{-3}$) | PM$_1$ (µg m$^{-3}$) | PM$_{2.5}$ (µg m$^{-3}$) | PM$_{10}$ (µg m$^{-3}$) | O$_3$ (ppb) | Org (µg m$^{-3}$) | SO$_4^{2-}$ (µg m$^{-3}$) | NO$_3^-$ (µg m$^{-3}$) | LDSA ratio | RH (%) | WS (m s$^{-1}$) | T (°C) |
|---|---|---|---|---|---|---|---|---|---|---|---|---|---|
| OPC.1 | 18.6±13.9 | 555±630 | 4.32±4.08 | 6.53±4.62 | 9.97±5.81 | 31.9±9.81 | 0.254±0.231 | 4.12E-02±5.42E-02 | 8.90E-02±1.15E-01 | 0.443 | 83.9±13.1 | 4.16±2.50 | 5.20±3.11 |
| OPC.2 | 9.64±1.90 | 233±32.8 | 2.56±0.72 | 5.61±1.58 | 10.7±2.97 | 38.6±1.34 | 0.142±0.082 | 2.98E-02±5.67E-02 | 1.64E-02±5.53E-03 | 0.847 | 65.1±10.5 | 7.1±1.01 | 7.16±1.53 |
| OPC.3 | 13.1±8.20 | 278±153 | 2.95±0.78 | 5.80±1.98 | 9.70±2.69 | 37.6±6.79 | 0.241±0.254 | 6.73E-02±6.25E-02 | 8.41E-02±1.54E-01 | 0.830 | 91.8±8.73 | 3.47±1.11 | 4.60±1.95 |
| OPC.4 | 11.5±7.15 | 281±191 | 2.51±1.55 | 4.84±3.20 | 8.33±5.35 | 36.5±5.17 | 0.192±0.235 | 4.53E-02±6.62E-02 | 1.08E-01±2.53E-01 | 0.536 | 83.5±11.5 | 4.37±2.09 | 6.26±2.73 |
| OPC.5 | 18.3±16.3 | 659±879 | 6.27±6.56 | 9.10±7.18 | 13.3±8.37 | 31.5±11.9 | 0.338±0.558 | 4.10E-02±6.49E-02 | 1.31E-01±2.62E-01 | 0.417 | 82.6±11.5 | 4.38±2.50 | 6.68±3.31 |
| OPC.6 | 8.58±6.72 | 197±155 | 2.85±1.12 | 5.96±2.51 | 10.3±4.30 | 40.0±4.69 | 0.116±0.152 | 3.50E-02±5.08E-02 | 3.50E-02±1.18E-01 | 0.588 | 81.2±12.3 | 4.87±2.07 | 6.42±2.89 |
| **Average** | **15.9±13.7** | **498±673** | **4.53±4.93** | **7.11±5.61** | **11.0±6.94** | **33.6±9.95** | **0.252±0.403** | **4.19E-02±6.05E-02** | **1.00E-01±2.08E-01** | **0.499** | **83.1±12.3** | **4.37±2.37** | **6.05±3.11** |

| | NO$_2$ (ppb) | BC (ng m$^{-3}$) | PM$_1$ (µg m$^{-3}$) | PM$_{2.5}$ (µg m$^{-3}$) | PM$_{10}$ (µg m$^{-3}$) | O$_3$ (ppb) | Org (µg m$^{-3}$) | SO$_4^{2-}$ (µg m$^{-3}$) | NO$_3^-$ (µg m$^{-3}$) | LDSA ratio | RH (%) | WS (m s$^{-1}$) | T (°C) |
|---|---|---|---|---|---|---|---|---|---|---|---|---|---|
| SMPS.1 | 16.0±14.9 | 485±852 | 3.35±2.64 | 5.70±3.89 | 9.52±6.05 | 32.2±10.3 | 0.215±0.300 | 3.06E-02±4.80E-02 | 5.47E-02±7.76E-02 | 0.331 | 85.1±10.7 | 4.1±2.70 | 5.53±3.06 |
| SMPS.2 | 16.8±12.0 | 406±539 | 2.70±1.57 | 5.11±2.33 | 8.91±3.75 | 32.9±8.10 | 0.132±0.156 | 2.53E-02±4.11E-02 | 2.56E-02±4.31E-02 | 0.501 | 83.2±9.71 | 3.74±1.67 | 4.64±2.86 |
| SMPS.3 | 4.38±2.91 | 88.1±62.2 | 2.64±1.62 | 5.57±3.62 | 9.26±5.87 | 41.6±3.24 | 0.062±0.063 | 3.74E-02±5.75E-02 | 2.07E-02±7.15E-02 | 0.555 | 80.1±8.93 | 7.19±2.48 | 7.43±2.72 |
| SMPS.4 | 14.3±12.3 | 452±592 | 3.77±2.56 | 6.71±3.75 | 11.1±5.67 | 35.6±9.32 | 0.249±0.306 | 4.68E-02±6.27E-02 | 8.12E-02±1.53E-01 | 0.499 | 79.4±13.9 | 4.74±2.38 | 6.97±2.62 |
| SMPS.5 | 29.8±17.2 | 1389±838 | 17.95±7.89 | 21.1±8.08 | 25.1±7.95 | 16.1±10.6 | 1.066±0.562 | 1.41E-01±7.58E-02 | 5.74E-01±3.60E-01 | 0.833 | 93.9±7.49 | 2.6±1.63 | 4.90±2.94 |
| SMPS.6 | 13.2±10.8 | 340±395 | 2.68±1.58 | 5.23±3.12 | 9.12±5.42 | 36.0±6.54 | 0.164±0.189 | 2.93E-02±4.31E-02 | 3.86E-02±7.17E-02 | 0.467 | 81.0±12.7 | 4.73±2.11 | 6.1±3.11 |
| **Average** | **15.1±13.2** | **460±649** | **4.12±4.72** | **6.78±5.48** | **10.8±6.90** | **33.8±9.84** | **0.280±0.403** | **4.61E-02±6.40E-02** | **1.07E-02±2.23E-01** | **0.499** | **82.8±12.4** | **4.41±2.42** | **5.95±2.99** |



**Table 3: Cluster relationships between OPC and SMPS clusters.**

| OPC/SMPS | SMPS.1 | SMPS.2 | SMPS.3 | SMPS.4 | SMPS.5 | SMPS.6 | Total OPC |
|---|---|---|---|---|---|---|---|
| OPC.1 | 48 | 30 | 9 | 71 | 13 | 66 | 237 |
| OPC.2 | 1 | 3 | | 5 | | 3 | 12 |
| OPC.3 | | 15 | | 2 | 4 | 2 | 23 |
| OPC.4 | 25 | 27 | 6 | 52 | 19 | 50 | 179 |
| OPC.5 | 24 | 26 | 17 | 39 | 40 | 38 | 184 |
| OPC.6 | 7 | 25 | 9 | 28 | 3 | 25 | 97 |
| Total SMPS | 105 | 126 | 41 | 197 | 79 | 184 | 732 |



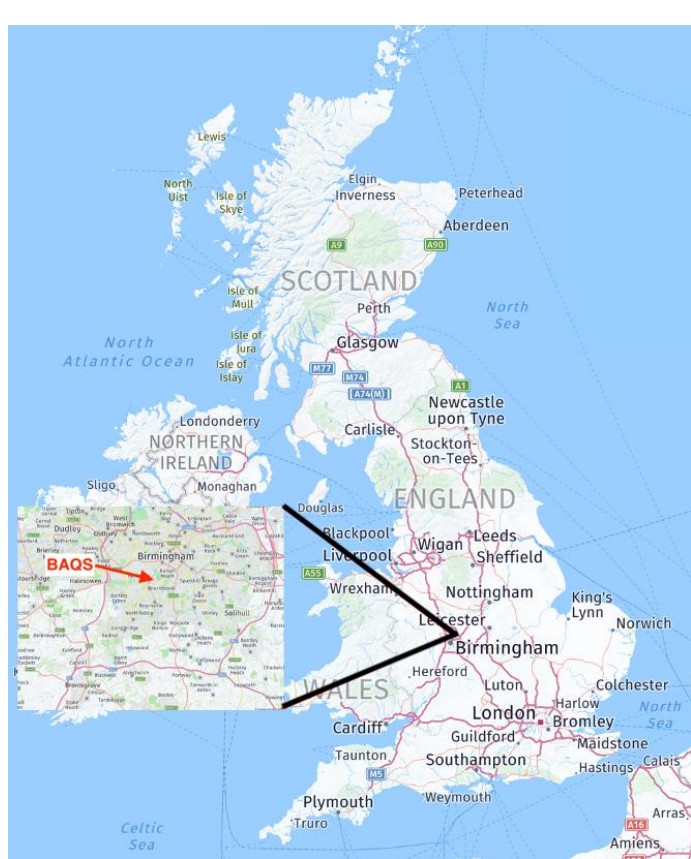

**Figure 1: Map of the location of the Birmingham Air Quality Supersite (BAQS) site in the U.K. (Map by ©HERE).**





Figure 2: Frequency and diurnal occurrence of the clusters formed by the OPC data.





**Figure 3: Frequency and diurnal occurrence of the clusters formed by the SMPS data.**





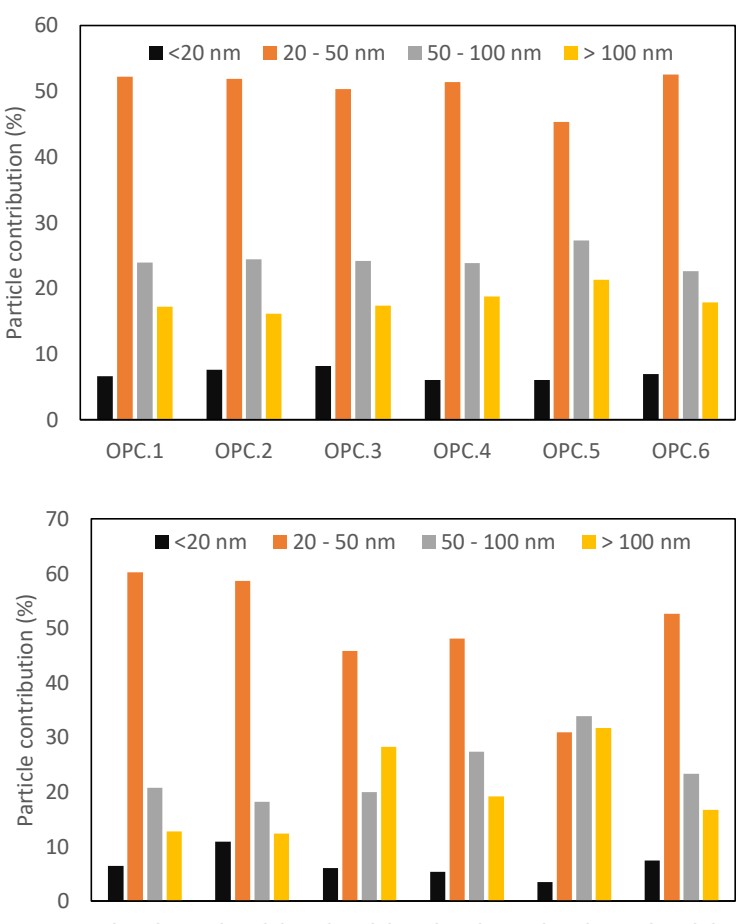

**Figure 4: Particle contributions in the range 12 – 550 nm, for the clusters formed using the OPC data (top) and the SMPS data (bottom).**





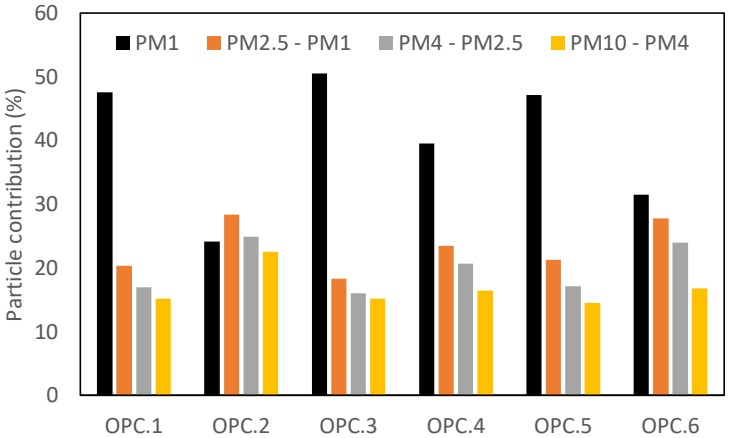

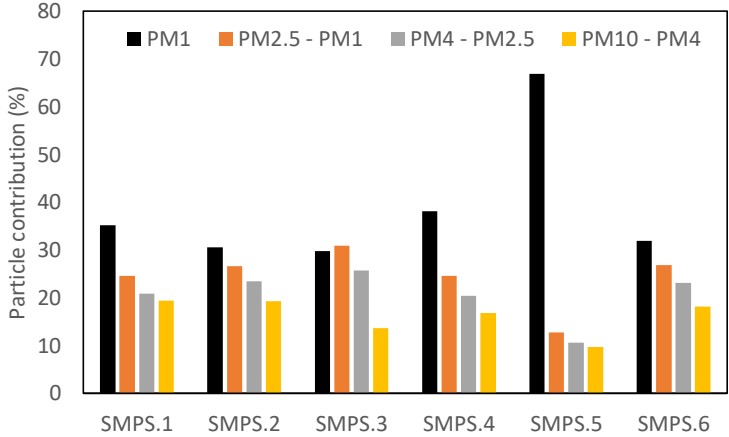

**Figure 5: Particle contributions up to 10 μm, for the clusters formed using the OPC data (top) and the SMPS data (bottom).**

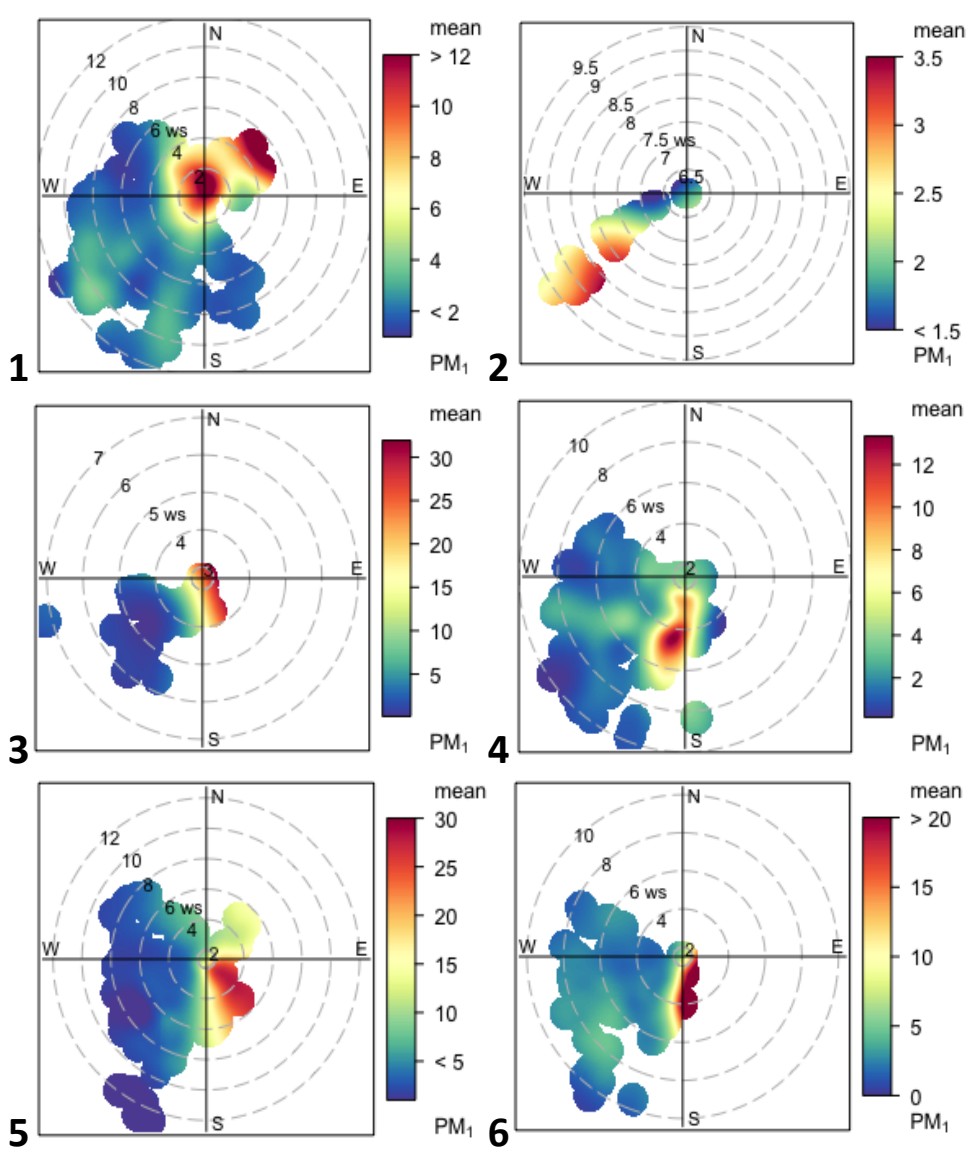

**Figure 6: PM$_1$ polar plots of the clusters formed by the OPC data.**





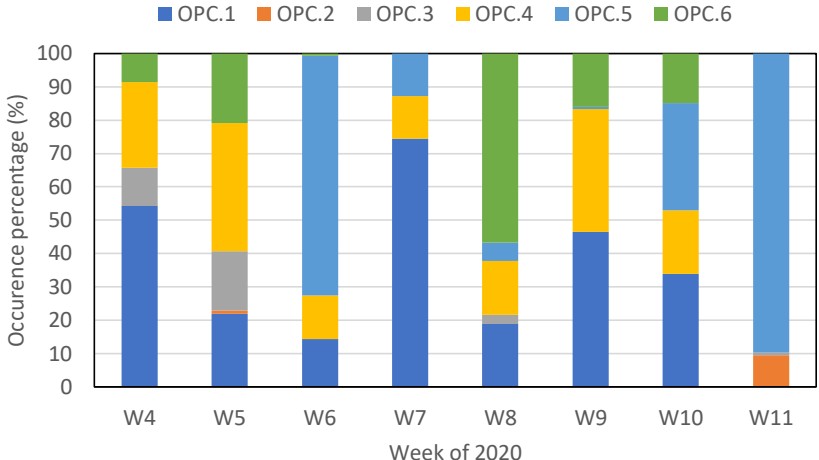

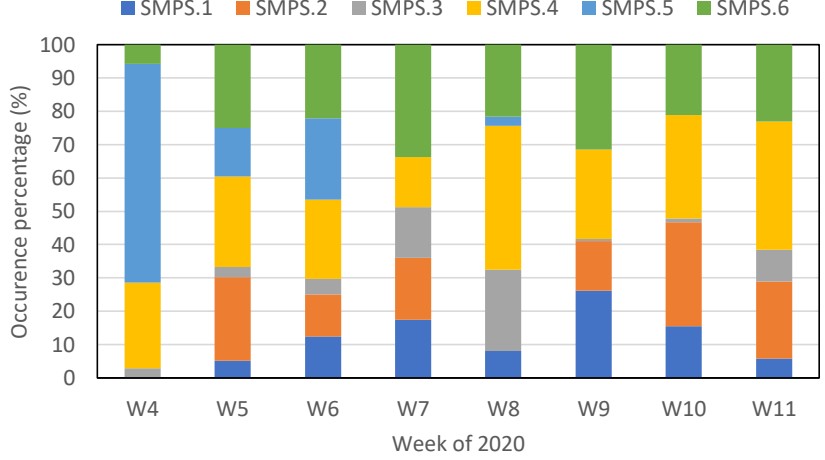

**Figure 7: Cluster percentage contribution per week (week number refers to week of year 2020).**