# Peer review of "Assessing the sources of particles at an urban background site using"

_Atmospheric Measurement Techniques, 2021_

## Author Comment (AC2)

**We thank the reviewers for their time and excellent insights which have improved the manuscript. We respond to all of the reviewers' points below.**

**Referee #1**

This manuscript presents an analysis to examine the ability of a low-cost optical particle counter (OPC) to separate atmospheric aerosol sources and conditions. The authors use k-means clustering on a low-cost OPC and a regulatory-grade scanning mobility particle sizer (SMPS). Given the different particle size ranges measured by the SMPS and the OPC, their performance differs by the sources and the temporal resolution they are able to untangle. Unsurprisingly, the OPC has limited success to separate the sources of smaller particles and higher temporal variability (e.g., diurnal variation). SMPS-based source identification performed well for sub-micron size range and was consistent with existing literature. As the authors themselves mention "the study of SMPS data with k-means clustering is far superior at separating complex pollution sources within urban environments in which the variation of very small particles is crucial for identifying particle and emission sources". However, the low-cost OPC based clustering performed well for particles in the 1–10 μm range and can have applications in regions and periods where coarse-mode particles are dominant such as dust storms, marine aerosol, bioaerosols, and other natural/resuspension sources.

I think the importance and quality of this manuscript warrants its publication in Atmospheric Measurement Techniques.

(Page 6, line 149) "…as increased concentrations of semi-volatile compounds are usually associated with anthropogenic sources, especially in the urban environment (Harkov, 1989; Schnelle-Kreis et al., 2007)."Requires more recent references — preferably based on online measurements.

**Response:** More recent references with measurements were added: Mahbub et al., 2011 and Xu and Zhang, 2011.

Does mixing layer height play an important role in some of the clusters? The authors can include summary mixing layer height for the existing clusters in Table S1 (Hourly MLH Reanalysis data can be obtained from ECMWF's ERA5).

**Response:** This was a nice idea – thank you for suggesting. The results for the PBL height were added at the Table 2. Its variation and importance in the clusters formed is discussed in several parts of the manuscript.

It seems that new particle formation was not observed during the measurement period. Is this consistent with the region and the measurement period? Please cite accordingly.

**Response:** A mention for the lack of NPF events and the general trends in the greater area (as recent studies for Birmingham do not exist) along with references for this were added.

Some relevant papers for this manuscript that can be cited:

- McGreggor et al. (1995), Synoptic Typing and its Application to the Investigation of Weather Air Pollution Relationships, Birmingham, United Kingdom, https://link.springer.com/article/10.1007/BF00867281 (Discuss the relationship of meteorology, air pollution, and use clustering analysis)
- Hagan and Kroll (2020), Assessing the accuracy of low-cost optical particle sensors using a physics-based approach, https://doi.org/10.5194/amt-13-6343-2020
- Brines et al. (2015), Traffic and nucleation events as main sources of ultrafine particles in high-insolation developed world cities, https://doi.org/10.5194/acp-15-5929-2015. (SMPS-based k-means paper)

**Response:** All suggested references were added to the text.

Minor comment: The figure captions for both the main manuscript and the supplement should be more descriptive.

**Response:** Figure captions were updated to provide clearer and more descriptive information for their figures.

Referee #2

This paper examines the ability of a low-cost OPC sensor in source apportionment by means of the k-means technique. This is conducted on a 48 days field campaign at an urban background site in Birmingham, UK. Furthermore, the outcome of the OPC analysis has been compared with those from a regulatory grade instrument (SMPS). Ancillary data from different low-cost and regulatory grade instruments ($NO_2$, $O_3$, $SO_4^{2-}$, $NO_3^-$, PM, organic content, BC, LDSA and meteorological parameters) have been acquired and used for the comparative study. Recently, there has been a large increase in the interest in low-cost monitoring sensors and this paper adds to the growing number of publications in the field, focusing the attention on the application of source apportionment to a low-cost OPC. The paper is useful, and it provides some good results, but a more robust comparison with PM source apportionment from the Fidas 200E should be conducted, as the OPC and SPMS have different size spectrum and a small overlap of size distribution. The information provided will be of much use to researchers in the future as they try to apply source apportionment techniques to low-cost sensor systems. Furthermore, although several works have been cited, the paper lacks in providing details, also from a mathematical point of view, of the k-means technique and its application. K-means technique is used for clustering of particle composition and is the core method of the paper and should be better outlined and described, allowing its reproduction by fellow scientists. Finally, there is a need for better indication in the text of cross-references to tables and figures when carrying out detailed scientific assessments and conclusions.

The paper should be published but the following comments should be taken into account and revisions should be made where necessary.

**Specific comments**

Which are the air flow rate and the wavelength of the laser beam of the OPC? (line 126)

**Response:** Flow rate and wavelength of the laser beam information added

Need to define PMx (line 129)

**Response:** PM sizes were added

How is the particle concentration calculated? Which are the values of the assumed parameters (e.g. particle density, etc)? (line 129)

**Response:** Additional information were added as requested

How are the larger coarse particles removed from the air stream of the OPC?

**Response:** The larger coarse particles are not deliberately removed from the air stream. The very large particles will be lost to impaction in the tubing prior to the OPC.  Any particles larger than 10 um will not be measured via the Mie scattering approach of the OPCs.  This information was added to the text.

High concentration of semi-volatile should be associated to a lower $LDSA_{ratio}$ since the LDSA after the catalytic stripper is expected to be lower than before. (line 150)

**Response:** The association of semi-volatile compounds to lower LDSA was added.

Which is the air flow rate of the pump? (line 160)

**Response:** Air flow rate information was added to the text

How is ozone calculated from oxidizing gas sensor $O_x$?

**Response:** Information about the calculation of $O_3$ from the $O_x$ measurements was added to the text.

A more in-depth description of how k-means clustering technique is applied to the acquired data must be provided, as well as the Dunn Index and the Silhouette width (Par. 2.2)

**Response:** The clustering technique is further explained and references are cited for each metric as well the k-means clustering for further reading. Additionally, the R libraries containing the functions used for such an analysis were added in the text.

What does "minimum and maximum cluster" mean? (line 196)

**Response:** The description of Dunn metric has been reworded to "The Dunn Index provides a measure of the ratio of the minimum cluster separation to the maximum (providing a metric of the compactness and separation of the clusters formed within the space)"

Clearly describe the sentence "a six-cluster solution was independently suggested for […]". What does "suggested" mean? Are you talking about convergence to a global optimum? (line 201)

**Response:** An explanation was added: *"the solution for which both metrics were maximum"* to justify the selection

A more robust comparison for the low-cost system is the application of the k-means to the Fidas 200E PM data (line 202).

**Response:** Clustering of FIDAS data would greatly limit the variation of the results from the regulatory grade instruments as out data set only comprises of 4 size bins. Thus, the note *"Though the clustering process could be applied for the FIDAS data, which are comparable in size range, it was not performed in this study."* was added to clarify the choice made.

Describe the content of Table 2 (line 208)

**Response:** A brief description of the results presented in table 2 was added

Why the occurrence of the OPC-1 during night-time is not so evident in figure 2? (line 235-line351)

**Response:** As evident throughout the analysis the OPC did not provide clear separation of the diurnal variation among the clusters. As a result, any diurnal variation of the OPC clusters is limited (which is pointed at the Conclusions as well). To "soften" the result presented in the text (at the spots pointed) the words "slightly" and "a somehow higher frequency" were added.

Why the occurrence of the OPC-6 during daytime is not so evident from figure 2? (line 267)

**Response:** The description of the diurnal variation of OPC.6 was removed as it was not correct as well as not explaining the highest $O_3$ concentrations. Instead the text was updated to *"Its association with cleaner conditions (lower concentrations of the pollutants with available data) probably explains the highest $O_3$ concentrations."* Which provides a more valid justification for the high concentrations of $O_3$.

Figure 4 – How is it possible that OPC distinguishes particles smaller than its lower bound range of 0.35 um? (line 337 and figure 4).

**Response:** In this analysis, for the groups formed by the OPC the SMPS data were considered to provide the variation at smaller sizes (trying to see whether the clustering of larger particles from the OPC may have a visible result even at sizes that it cannot measure – i.e. whether the footprint of sources of smaller particles is visible at the greater sizes via simultaneous particle emissions in different sizes). In order to clarify this the text was updated to *"As expected, the SMPS is more capable in separating different cluster profiles at the size range smaller than*

*500 nm, a size range in which the cluster profiles (using the SMPS data) formed by the groups from the OPC are almost uniform (fig. 4)"*

Clearly argue the following sentence "The clusters formed using the OPC data were also better associated with different sources of PM1 (fig. 6), compared to those deriving from the SMPS data (fig. S7)" (line 341)

**Response:** Added a clarification for the meaning of this sentence (line 399)

Which table shows that the BC and ozone concentrations are high for OPC.3 group? (line 426)

**Response:** The results included in the analysis were only from the regulatory grade instrument. Due to instrument error though, only 2 hours had available data for these chemical components, biasing the results. Thus, the note *"the results provided in table 2 for the OPC.3 are only from 2 hours of data that were available from the regulatory grade instrument"* was added to point the discrepancy found, as well as a note that the BC results from the low-cost sensor are not included.

Please, consider inclusion of the following recent papers:

- Dall'Osto et al. (2012) Urban aerosol size distributions over the Mediterranean city of Barcelona, NE Spain, Atmospheric Chemistry and Physics
- Morawska et. Al. (2018), Applications of low-cost sensing technologies for air quality monitoring and exposure assessment: how far have they gone? Environment International
- Shindler (2020), Development of a low-cost sensing platform for air quality monitoring: application in the city of Rome, Environmental Technology

**Response:** All suggested papers were included in the text.

**Technical comments**

The captions must fully describe figures and tables

**Response:** All figure and table captions have been updated to provide clearer and more descriptive information for their figures.

A better representation of figures S2 and S5 must be provided

**Response:** The figures presented include information about the particle number concentration (and the quartiles) and particle volume (important as it provides some information for the variation of larger particles from the OPC, which appear as flat lines when their number is considered). We cannot think of a better way to present this information.

The A33 Aetholometer is not described in the text (Table 1)

**Response:** A description of the A33 Aetholemeter was added

Specify also in text the manufacturer of the EC sensors (line 160)

**Response:** The electrochemical sensors (EC) are a part of the BOCS for which the manufacturer was added in the text.

Specify also in the text the manufacturers of the SPMPS and of the ACSM instruments (line 170)

**Response:** The manufacturers of both instruments were added in the text.

A picture of the Birmingham Air Quality supersite with all the instruments should be provided (line 175)

**Response:** A photo of the low-cost sensors used at BAQS was added in the SI.

Need to specify the abbreviation "Org" reported in Table 2 when it is first defined in the text (line 173)

**Response:** "Org" was changed to organic content in table 2

Fassociated is associated (line 299)

**Response:** Corrected

Figure S1 and S7 - No unit of measurement is reported on the polar plots

**Response:** Units were added in the caption of the figures

Figure 7 - Using the same colours for OPC and SPSM clusters is misunderstanding, since the groups do not correspond to each other. A better representation should be used

**Response:** A pattern was added for the SMPS results to differentiate them from those of the OPC.

Table S2 – Verify units of measurement of PM concentrations. What are the statistics reported in the table?

**Response:** Units of PM measurements are corrected. Caption of Table S2 was updated to clearly state what is presented

2 occurring day is probably 12/3? (line 403)

**Response:** May was corrected to March

NO3. (line 404)

Response: Corrected to $NO_3^-$

---

## Author Response (AR2)

Response to technical comments

Dear Authors of AMT-2021-11,
I'd like to thank you and the reviewers for their contributions, which in my opinion greatly helped the quality of the original manuscript. I think you satisfactorily addressed all points raised by the reviewers, and recommend the paper for publication in AMT.

Thank you for the kind comments.

The language was improved greatly too from the very first submission, but I encourage you to revise it yet another time, for better readability.

We have reviewed the document again and have improved the English in a few places.

I have some small technical points I'd like you to solve before publication in AMT, listed below:
L133- 1.5 +/- 0?

We changed the sentence from "Particle mass loadings ($PM_1$, $PM_{2.5}$ and $PM_{10}$) are then calculated from the particle size spectra and concentration data, assuming a particle density and refractive index (default density is 1.65 g/ml and refractive index is 1.5+i0)." to "Particle mass loadings ($PM_1$, $PM_{2.5}$ and $PM_{10}$) are then calculated from the particle size spectra and concentration data, assuming a particle density and refractive index (default density is 1.65 g/ml and complex refractive index is 1.5+i0)." to make clear that i refers to the complex part of the refractive index.

L136, L140: remove "which" and provide-provides

Changed

L143 Nd 1.1 in superscript: this is not very clear, I think you should express it more explicitly.

We have extended the section to the following:

"Two Naneos Partectors (Naneos Particle Solutions GmbH) provide the lung deposited surface area metric (LDSA, $\mu m^2/cm^3$) in the particle diameter range 10 nm to 10 $\mu$m. In general, the instrument charges particles with an efficiency proportional to the particle diameter to the power of 1.1 ($d^{1.1}$) and is independent of particle composition. The particle number concentration (*N*) is also provided for all particles, resulting in a $Nd^{1.1}$ metric that can be correlated to LDSA."

L230-231: I have the impression that adding exactly the explanation you provided to Reviewer#2 in your responses, would further help the explanation in the paper.

We have enhanced the explanation to "Though the clustering process could be applied for the FIDAS data, which are comparable in size range, it was not performed in this study because of the limited size bin data of the FIDAS instrument."

L395: extend-extent.

changed